# Aligning to the teacher: multilevel feature-aligned knowledge distillation

Yang Zhang[1],*, Pan He[1],*, Chuanyun Xu[1], Jingyan Pang[1], Xiao Wang[1], Xinghai Yuan[1], Pengfei Lv[2] and Gang Li[2]

[1] School of Computer and Information Science, Chongqing Normal University, Chongqing, China
[2] School of Artificial Intelligence, Chongqing University of Technology, Chongqing, China
* These authors contributed equally to this work.



## ABSTRACT

Knowledge distillation is a technique for transferring knowledge from a teacher's (large) model to a student's (small) model. Usually, the features of the teacher model contain richer information, while the features of the student model carry less information. This leads to a poor distillation effect in the process of knowledge transfer due to insufficient feature information and poor generalisation ability of the student model. In order to effectively reduce the feature differences between teacher–student models, we propose a Multilevel Feature Alignment Knowledge Distillation method (MFAKD), which includes a spatial dimension alignment module and a multibranch channel alignment (MBCA) module. The upsampling operation enables the student feature map to align with the teacher feature map in spatial dimensions, allowing students to learn more comprehensive teacher features. Moreover, MBCA achieves the alignment of the student feature maps and teacher feature maps on channels, which can transform the student features into a form more similar to the teacher features. Subsequently, the student model uses the discriminative classifiers from the pretrained teacher model to perform the student model inference. In summary, the student model can utilize their strengths and surpass the teacher model. The method was validated on the CIFAR-100 dataset and obtains state-of-the-art results. In particular, the classification accuracy of the student model WRN-40-2 exceeds that of the teacher model ResNet-8×4 by almost 2%. The method also demonstrated excellent performance on the Tiny ImageNet dataset.

## INTRODUCTION

Deep neural networks (DNNs) have made significant progress in the field of computer vision image recognition, but deep neural networks have a large number of parameters and are difficult to deploy; this often leads to considerable overhead in both storage and computation. To solve this problem, *Hinton, Vinyals & Dean (2015)* proposed the knowledge distillation (KD) method, which passes knowledge from the teacher model to the student model by minimizing the output difference between the teacher model and the student model. *Hu et al. (2022)* believes that the KD method is simple, easy to implement, and widely used in various image recognition tasks.

Corresponding authors
Chuanyun Xu, xcy@cqnu.edu.cn
Gang Li, ligang@cqut.edu.cn

The KD approach allows the student model to learn from the final output of the teacher's model, disregarding the learning process in between. Gou et al. (2021) proposed that later research focused on students' learning processes, which is considered feature-based knowledge extraction. For example, Chen et al. (2021b) proposed the ReviewKD method, which enables the student model to review the knowledge they have already learned through a knowledge review mechanism, which is the fusion of shallow intermediate features; Shallow and deep features contain different semantic and spatial information. Zhang et al. (2023) proposed the RAIL-KD method, which involves randomly selecting intermediate layers of teacher and student models in each training round and establishing a map for knowledge transfer. Therefore, the student model can learn more comprehensive knowledge from the teacher model by means of random instruction. Chen et al. (2021a) proposed the SemCKD method, which achieves cross layer knowledge transfer through semantic calibration; The semantics of different layers in different neural networks may vary, and semantic mismatches in artificial layer associations can lead to performance degradation caused by negative regularization. Chen et al. (2022) proposes the SimKD method, which uses a single projector to transform student features and reuses pre-trained teacher classifiers for student inference. However, the simple alignment of student characteristics with teacher characteristics limits the ability of student models to learn freely. Therefore, the design an appropriate projector is a problem warranting further research.

Traditional Chinese education uses the term "follow others at every step" to criticize the teaching style; here, students strictly imitate the teacher's behavior. In traditional Chinese education, "following others at every step" is rejected because students may rely too much on the teacher's thinking and struggle to flexibly build on their own knowledge for a deeper understanding of new concepts. This overreliance on the teacher's framework often limits students' ability to surpass their instructors' grasp of the subject matter. Therefore, students should not only accept the knowledge imparted by the teacher but also, more importantly, develop the ability to think about the problem in multiple ways, as shown in the intuitive schematic diagram in Fig. 1. In knowledge distillation, the student model traits are learned from the teacher traits from multiple perspectives by designing a multibranch projector.

In this study, we propose a Multilevel Feature Alignment Knowledge Distillation method (MFAKD). This method employs upsampling operations to align student features with teacher features, enabling the student to learn a broader range of teacher features when the spatial dimensions of student features are smaller than those of the teacher. In the channel dimensions, the goal of MFAKD is to create multiple mapping branches between the teacher features and the student features such that the student features can be aligned with the teacher features from different perspectives. The feature projectors can adjust student features and transform them into a form that is more similar to teacher features. Multilevel feature projectors helps to enhance the diversity of student features, and integrating multilevel feature projectors not only enhances the generalisation ability of student features but also effectively alleviates the problem of teachers' overconfidence. A large number of experimental results have shown that the Multilevel Feature Alignment

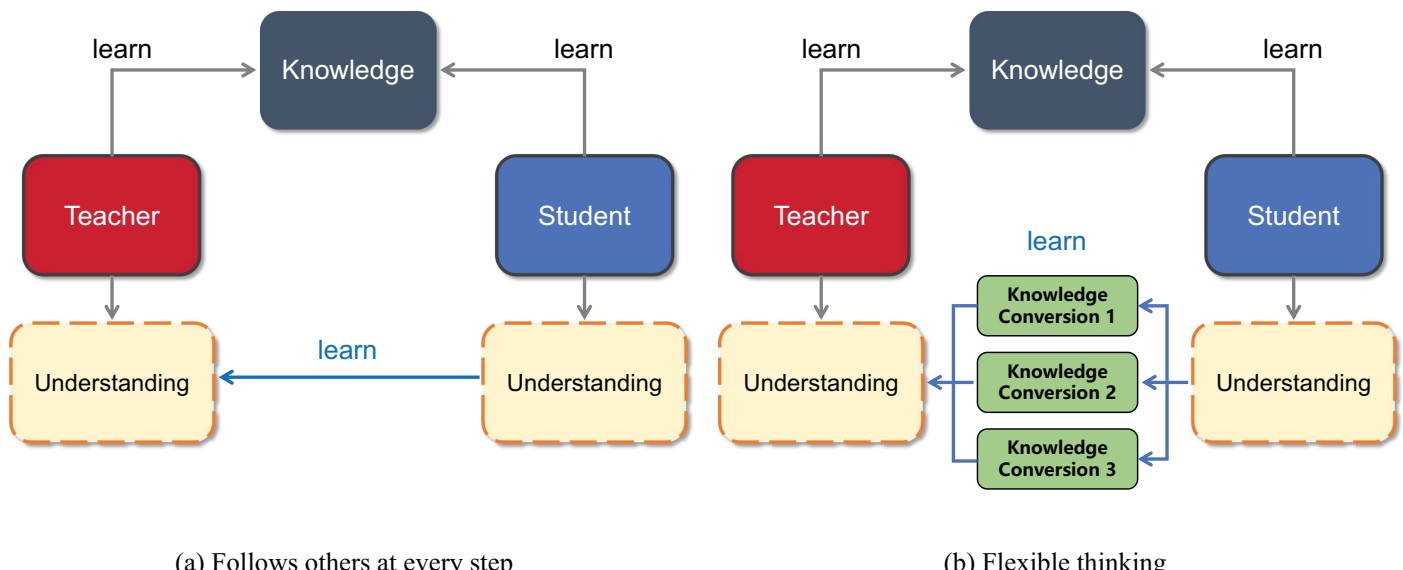

(a) Follows others at every step          (b) Flexible thinking

**Figure 1** **Visualization of the ideas presented in this article.** (A) Shows that students learn "follow others at every step" from the teacher, and (B) shows that students understand what the teacher teaches from multiple perspectives.

Knowledge Distillation (MFAKD) method is effective. This study contributes the following:

- We have found that the traditional forced matching way between student feature maps and teacher feature maps has drawbacks, which can lead to a lack of independent thinking ability among students, thereby limiting the effectiveness of students obtaining knowledge from teachers.
- We propose a Multilevel Feature Alignment Knowledge Distillation method, which can effectively adjust student features and transform them towards a more similar direction to teacher features, thereby enhancing the diversity of student features. Moreover, integrating multilevel feature projectors can not only enhance students' ability to generalise features, but also significantly alleviate teachers' overconfidence problem.
- Numerous experimental designs have demonstrated that the student model can outperform the teacher model when student features are simultaneously aligned with teacher features in both spatial and channel dimensions. On the validation set of the CIFAR-100 dataset, the classification accuracy of the student model WRN-40-2 outperforms that of the teacher model ResNet-8×4 by nearly 2%. On the Tiny ImageNet dataset, the MFAKD method also demonstrated excellent performance.

## RELATED WORK

*Hinton, Vinyals & Dean (2015)*, *Wang & Yoon (2021)*, and *Gou et al. (2021)* believes that KD is a model compression technique in deep learning, which achieves model compression by training on large and complex models (teacher models) and then transferring their knowledge to smaller and simpler models (student models).

*Hinton, Vinyals & Dean (2015)*, *Ba & Caruana (2014)*, and *Chen et al. (2020)* believes that the original knowledge distillation technique uses the output distribution of the model as knowledge, and the reduced output of the teacher model plays a crucial role in improving the performance of the student model, as the reduced output generates relationships between different categories and has beneficial generalisation effects during student training.

*Gou et al. (2021)* and *Wang & Yoon (2021)* believes that in order to obtain better distillation results, it is necessary to further investigate the gap between the student model and the teacher model. Usually, it is recommended to use the intermediate features of the model as knowledge. In previous studies, most models used different ways to express intermediate features, such as feature-based knowledge extraction first proposed by *Adriana et al. (2015)* and following studies *Ahn et al. (2019)*, *Heo et al. (2019)*. There are also some studies on *Tian, Krishnan & Isola (2019)*, *Xu et al. (2020)* that establish sample relationships through comparative learning. The core idea of ReviewKD is to establish a knowledge review mechanism from deep to shallow layers of neural networks for distillation. Both this study and the ReviewKD method utilize multilevel features, with the difference being that ReviewKD achieves multiscale feature distillation through shallow and deep features of the network, while this study projects the last layer of network features into multilevel features for distillation. Another effective way for feature distillation is self-supervised knowledge, such as the HSAKD proposed by *Yang et al. (2021)*. HSAKD combines the label space of the original classification task and the self supervised task into a joint label space by introducing a self supervised enhancement task to generate richer knowledge, and adds auxiliary classifiers to the hierarchical intermediate feature maps of the teacher and student networks for one-to-one probabilistic knowledge distillation, thereby improving the representation ability and classification performance of the student network. In the past two years, some researchers, *Ahn et al. (2019)* and *Chen et al. (2021b)* have utilised the intermediate features of the teacher model through cross-layer connections between the designer student model architecture. Currently, *Chen et al. (2022)*'s research focuses on using teacher classifiers for student model inference. This method introduces a projector before reusing the teacher classifiers, and the projector caused the student features to be closer to the teacher features; however, research on projectors in knowledge distillation is limited, and a method to design a more optimal projector is not yet available. Therefore, this study investigates this problem.

The existing knowledge extraction methods using projectors, such as contrastive representation distillation (CRD) proposed by *Tian, Krishnan & Isola (2019)*, softmax regression representation learning (SRRL) proposed by *Yang et al. (2020)*, and comprehensive, interventional distillation (CID) proposed by *Deng & Zhang (2021)*, typically use convolutional or linear projectors to transform features, but their impact on projectors has not been thoroughly explored. At present, *Chen et al. (2023)* proposes the Projector Ensemble Feature Distillation (PEFD) method, which improves performance by distilling the projector features of the final result and integrating projector ideas. Ensemble learning requires mixing multiple models together to predict the results; these depend on multiple models. The SimKD distillation method proposed by *Chen et al. (2022)* uses a

single projector to convert student features, but the single feature transformation is too "tough." To solve these two problems, the use of multibranching provides a more superior solution. This method combines multiple weak projectors in parallel to form a strong projector layer to increase the generalization ability of student features. In the standard distillation experimental environment, this method demonstrated good performance and achieved excellent results.

# MULTILEVEL FEATURE ALIGNMENT KNOWLEDGE DISTILLATION

The core concepts of multilevel feature alignment knowledge distillation are spatial dimension alignment and channel alignment, as shown in Fig. 2. Therefore, this section is mainly presented in terms of spatial dimension alignment and channel alignment.

## Spatial dimension alignment

In knowledge distillation, the spatial dimension of student feature maps is usually smaller than that of teacher feature maps. Therefore, this study uses an upsampling operation to align student features with teacher features. Upsampling operations can help increase the size of feature maps, thereby improving the spatial resolution of features. This helps the network better capture detailed information in images, thereby improving the performance of the model in handling complex tasks.

Let the spatial feature dimension of the student model be $H_s \times W_s$, and that of the teacher be $H_t \times W_t$. To upsample the feature map of the student model to match that of the teacher, let the upsampled student feature map be $M_{spaceAlign}$. The spatial feature alignment between the student model and the teacher model is shown in Formula (1). Specific upsampling methods will be discussed in subsequent experiments.

$$M_{\text{spaceAlign}} = rH_s \times rW_s = H_t \times W_t \tag{1}$$

where $r$ represents the upsampling ratio.

## Channel alignment

In this study, three branch mappings were created between the teacher–student channel to create a "bridge" from the student's features to the teacher's features, a methodology known as multibranch channel alignment (MBCA). MBCA has three parts in parallel: bottleneck, sandglass, and sandglass+, as shown in Fig. 3. They each use multiple convolutional kernels of different scales to extract different features. Thus, students can effectively understand teacher features from multiple perspectives.

**Bottleneck projector.** Bottlenecks are more economical in terms of the number of parameters and still maintain improved performance. It is processed by downscaling the student features *via* $1 \times 1$ convolution, which decreases the number of channels downscaled by half of the original. Then, by $3 \times 3$ convolution, the number of input channels is equal to the number of output channels, and finally, $1 \times 1$ convolution is used for upscaling, which restores the original number of channels. Standard batch normalization and ReLU activation are used after each convolutional layer of the

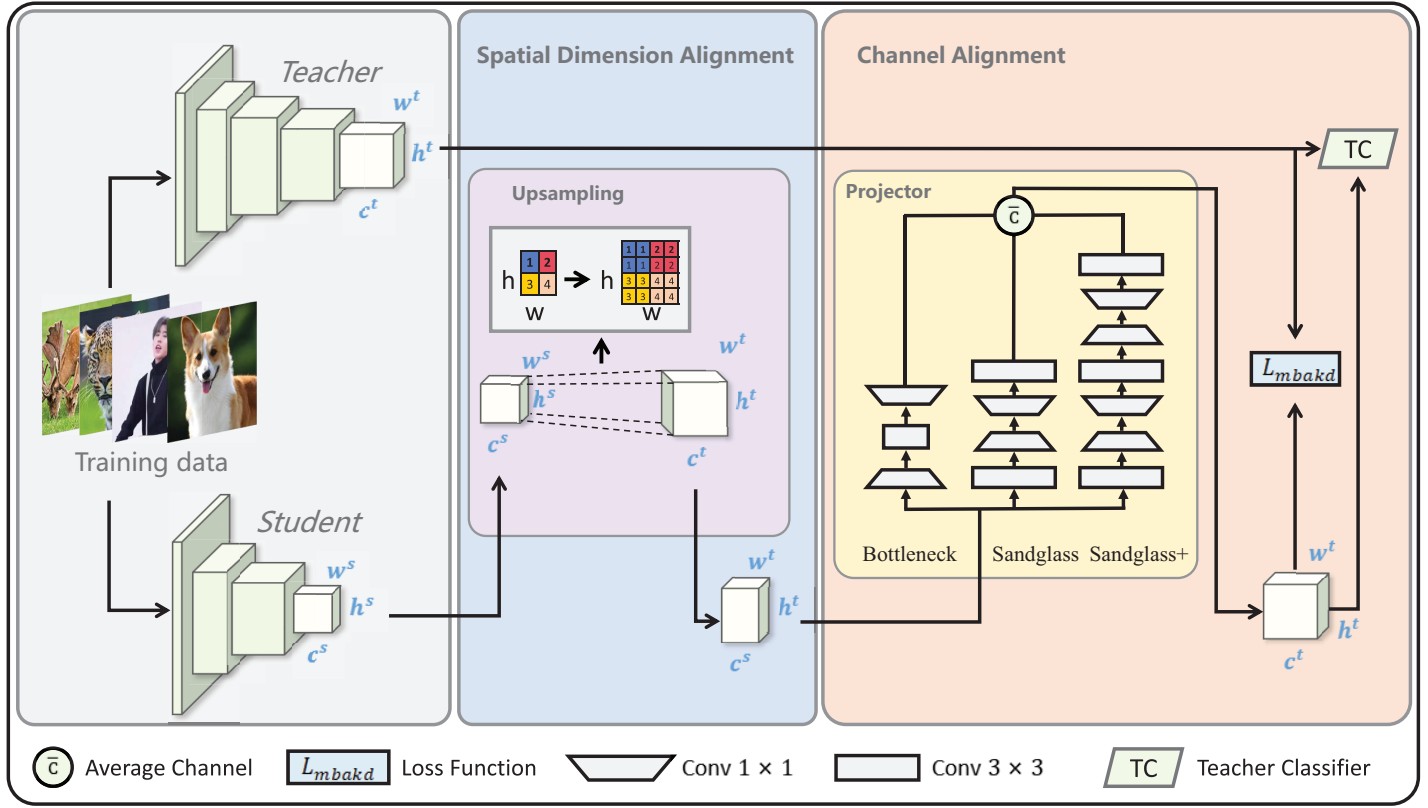

**Figure 2 Diagram of multilevel feature alignment knowledge distillation framework.** In the spatial dimension, student features are aligned to the teacher's spatial dimensions *via* upsampling. $w^S = w^T$ and $h^S = h^T$ after the spatial dimension alignment; the channel dimension is aligned with the channel dimension of the teacher using the multibranch projector. $c^S = c^T$ after the channel dimension alignment. Subsequently, the loss is computed by aligning the student features with the teacher features in both the spatial and channel dimensions and is finally classified using the teacher classifier TC.

bottleneck projector, as shown in Fig. 3A. The conversion process for the remaining two branches is the same as for the bottomleneck projector; to save space, the latter is omitted from the process introduction.

**Sandglass projector.** Sandglass is the superior projector structure, as shown in Fig. 3B. The bottleneck structure in Fig. 3A causes gradient confusion due to narrower feature dimensions, and it also weakens the ability of the gradient propagation across the layers; *Zhou et al. (2020)* believes that this affects the training convergence and the model performance. Sandglass retains more information and back-propagates more gradients to better optimize network training.

**Sandglass+ projector.** *Adriana et al. (2015)* proposed FitNet to train narrower and deeper student networks with wider and more complex teacher networks. The design of the projector may also follow the better design of narrower and deeper structures. Therefore, in this study, the Sandglass Projector is extended and named the Sandglass+ Projector, as shown in Fig. 3C.

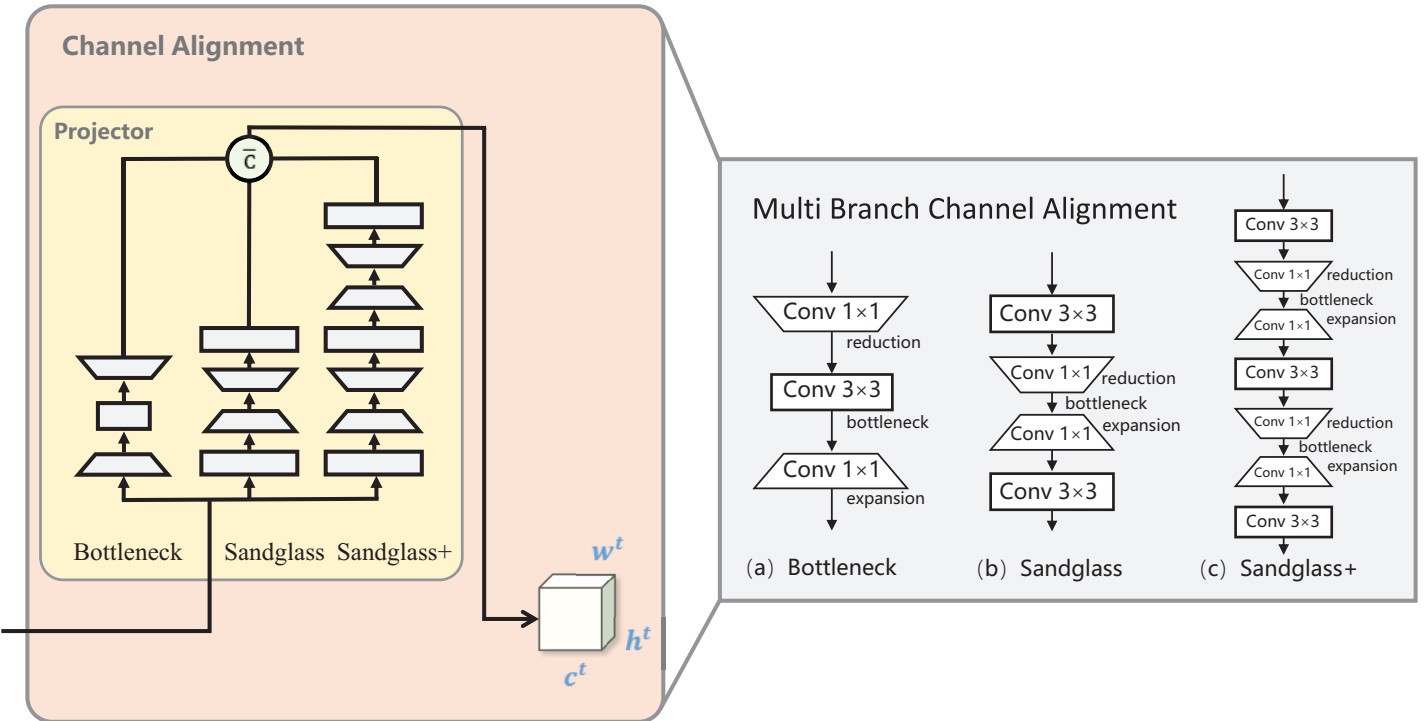

**Figure 3 Multibranch channel alignment.** The projector was designed to address the teacher–student channel dimension mismatch and the fact that the multibranch projector allows the student model to understand the teacher model knowledge from multiple perspectives. $c^S = c^T$ after the channel dimensions are aligned.

**Multibranch channel alignment projector.** MBCA constructs a strong projector by parallel connecting multiple weak projectors, which not only enriches the diversity of student features and enhances their generalization ability, but also effectively solves the problem of feature dimension mismatch between heterogeneous teacher–student models. Let the channel feature dimension of the student model be $C_s$, and that of the teacher be $C_t$. For the sake of brevity, let the feature calculation process of the Bottleneck Projector be denoted as $BP$, the calculation process of the Sandglass Projector be denoted as $SP$, and the calculation process of the Sandglass+ Projector be denoted as $SP^+$.

$$BP(C_s) = rC_s = C_t \tag{2}$$
$$SP(C_s) = rC_s = C_t \tag{3}$$
$$SP^+(C_s) = rC_s = C_t \tag{4}$$

where $r$ represents the scaling factor. For the sake of brevity, set $BP(f_s)$, $SP(f_s)$, and $SP^+(f_s)$ as $\frac{1}{n}\sum_{n}^{i=1} P_i(f^s)$. The loss function in this study is shown in Eq. (5).

$$Loss_{mfakd} = \alpha \left\| f^t - \frac{1}{n}\sum_{i=1}^{n} P_i(f^s) \right\|_2^2 \tag{5}$$

Among them, $\alpha$ is the weight of the loss function, $f^t$ represents the feature map of the teacher model, $f^s$ represents the feature map of the student model, $n$ is the number of

projectors, and $P(f^s)$ is the projected features of the student. Equation (5) utilizes upsampling and multibranch channel-aligned projectors to transform the student features and align them with the teacher features. Then, the integrated multibranch projected student features are obtained by taking the average. Finally, calculate the mean square error between the student features transformed by the multibranch projectors and the teacher features.

## EXPERIMENTS

In this section, multiple groups of combined teacher–student experiments are designed to verify the validity of MFAKD. Comparisons are made with state-of-the-art methods from recent studies using the CIFAR-100 dataset and the tinyImageNet dataset, and then the data is analyzed to demonstrate the validity of multibranch channel-aligned MBCA.

**DataSet and baselines.** This experiment uses the CIFAR-100 baseline image classification dataset using standard data enhancement (random cropping and level flipping), and all images are normalized by channel means and standard deviations. In this study, the FitNet, AT, SP, VID, OFD, CRD, SRRL, SemCKD, ReviewKD, SimKD, DKD, HSAKD, and SDD distillation methods are used for comparison, and all of these methods use the L2 loss function.

**Training details.** The experiments follow the training procedures of previous authors (*Tian, Krishnan & Isola, 2019*; *Chen et al., 2021a*; *Yang et al., 2020*), and the performance of all competitors are reported on a randomized set of 14 teacher–student model combinations. Specifically, the SGD optimizer was used for all data with a momentum of 0.9 Nesterov momentum. For the CIFAR-100 dataset and the tinyImageNet dataset, the total training period was set to 240 epochs, and the initial learning rate was set to 0.05, divided by 10 at rounds 150 epochs, 180 epochs, and 210 epochs. The MobileNet/ShuffleNet series learning rate was set to 0.01. The batch size was set to 64, and the weight decay was set to $5 \times 10^{-4}$, and Set the weight of the loss function $\alpha$ to 1.0. All experiments in this paper are averaged over three trials. On the CIFAR-100 dataset, when using an RTX 3080 GPU, the individual training time for the teacher model across various teacher–student model combinations is approximately 7–8 h, the individual training of the student model requires about 3–4 h, and training with the MFAKD method takes roughly 4–5 h. On the TinyImageNet dataset, with an RTX 3080 GPU, the individual training time for the teacher model in various teacher–student model combinations is around 3–4 h, the student model training takes approximately 1–2 h, and the MFAKD method only requires about 2 h for training.

### Comparison of test accuracy

As shown in Tables 1 and 2, the performance of Multilevel Feature Alignment Knowledge Distillation (MFAKD) is tested based on 14 different teacher–student combinations. It is compared with the methods proposed by authors (*Hinton, Vinyals & Dean, 2015*; *Adriana et al., 2015*; *Zagoruyko & Komodakis, 2016*; *Tung & Mori, 2019*; *Ahn et al., 2019*; *Tian, Krishnan & Isola, 2019*; *Yang et al., 2020*; *Chen et al., 2021a*; *Chen et al., 2022*). From

**Table 1 Top-1 test accuracy (%) of various knowledge distillation approaches on CIFAR-100.** The bold text indicates the best classification accuracy.

| Student | WRN-40-1 | ResNet-8×4 | ResNet-110 | ResNet-116 | VGG-8 | ResNet-8×4 | ShuffleNetV2 |
|---|---|---|---|---|---|---|---|
| Acc | 69.92 ± 0.17 | 73.09 ± 0.30 | 74.37 ± 0.17 | 74.46 ± 0.09 | 70.46 ± 0.29 | 72.09 ± 0.28 | 72.60 ± 0.12 |
| Teacher | WRN-40-2 | ResNet-32×4 | ResNet-110×2 | ResNet-110×2 | ResNet-32×4 | WRN-40-2 | ResNet-32×4 |
| Acc | 75.61 | 79.42 | 78.18 | 78.18 | 79.42 | 75.61 | 79.42 |
| KD | 72.01 ± 0.25 | 74.42 ± 0.05 | 76.25 ± 0.34 | 76.14 ± 0.32 | 72.73 ± 0.15 | 74.28 ± 0.15 | 75.60 ± 0.21 |
| FitNet | 72.36 ± 0.15 | 74.32 ± 0.08 | 76.08 ± 0.13 | 76.20 ± 0.17 | 72.91 ± 0.18 | 74.02 ± 0.29 | 75.82 ± 0.22 |
| AT | 72.55 ± 0.19 | 75.07 ± 0.03 | 76.67 ± 0.28 | 76.84 ± 0.25 | 71.90 ± 0.13 | 74.74 ± 0.11 | 75.41 ± 0.10 |
| SP | 71.89 ± 0.11 | 74.29 ± 0.07 | 76.43 ± 0.39 | 75.99 ± 0.26 | 73.12 ± 0.10 | 73.84 ± 0.10 | 75.77 ± 0.08 |
| VID | 72.64 ± 0.13 | 74.55 ± 0.10 | 76.17 ± 0.22 | 76.53 ± 0.24 | 73.19 ± 0.23 | 74.56 ± 0.17 | 75.22 ± 0.07 |
| OFD | 72.84 ± 0.17 | 74.65 ± 0.13 | 76.27 ± 0.12 | 76.83 ± 0.14 | 73.32 ± 0.11 | 74.76 ± 0.19 | 75.62 ± 0.17 |
| CRD | 72.88 ± 0.26 | 75.59 ± 0.07 | 76.86 ± 0.09 | 76.83 ± 0.13 | 73.54 ± 0.19 | 74.78 ± 0.30 | 77.04 ± 0.61 |
| SRRL | 72.57 ± 0.19 | 75.39 ± 0.34 | 76.75 ± 0.14 | 77.19 ± 0.09 | 73.23 ± 0.16 | 75.12 ± 0.17 | 76.19 ± 0.35 |
| SemCKD | 72.45 ± 0.18 | 76.23 ± 0.04 | 76.62 ± 0.14 | 76.69 ± 0.48 | 75.27 ± 0.13 | 74.85 ± 0.18 | 77.62 ± 0.32 |
| ReviewKD | 72.93 ± 0.22 | 75.63 ± 0.17 | 76.87 ± 0.16 | 76.90 ± 0.14 | 75.26 ± 0.18 | 74.75 ± 0.14 | 77.19 ± 0.15 |
| SimKD | 74.12 ± 0.15 | 78.08 ± 0.15 | 77.82 ± 0.15 | 77.90 ± 0.11 | 75.76 ± 0.12 | 76.75 ± 0.14 | 78.39 ± 0.27 |
| DKD | 74.81 ± 0.10 | 76.32 ± 0.14 | 77.09 ± 0.18 | 76.89 ± 0.11 | 75.20 ± 0.11 | 74.80 ± 0.16 | 77.07 ± 0.10 |
| HSAKD | 74.53 ± 0.12 | 77.26 ± 0.14 | 77.66 ± 0.10 | 77.32 ± 0.13 | 75.36 ± 0.17 | 74.96 ± 0.16 | 77.69 ± 0.10 |
| SDD | 73.22 ± 0.12 | 76.63 ± 0.10 | 76.77 ± 0.09 | 76.99 ± 0.13 | 75.51 ± 0.17 | 75.08 ± 0.23 | 77.88 ± 0.23 |
| MFAKD | **75.62 ± 0.16** | **79.90 ± 0.08** | **78.08 ± 0.11** | **78.19 ± 0.11** | **76.27 ± 0.13** | **77.39 ± 0.18** | **78.60 ± 0.14** |

Tables 1 and 2, MFAKD consistently outperforms SimKD on CIFAR-100, and the improvement is very significant in certain teacher–student model combinations, such as ResNet32×4 and ResNet8×4, WRN-40-2 and ResNet8×4, ResNet32×4 and WRN-40-2, and ResNet32×4 and ShuffleNetv2×1.5; in these combinations, the student model have completely surpassed the teacher model. After observation, it was found that the distillation effect is better when the spatial dimension of the student feature map is the same as that of the teacher feature map. When the spatial dimension of the student feature map is smaller than that of the teacher feature map, aligning the student feature map with the teacher feature map yields better distillation results. The best performing teacher–student model combination is WRN-40-2 and Resnet8×4. The testing accuracy of the student model exceeds that of the teacher model by nearly 2%. Among them, the channel number of the student model Resnet8×4 is twice that of the teacher model WRN-40-2, and the spatial dimension is the same as the teacher model. In addition, to help the network learn and optimize the model parameters for better performance, the MFAKD method attempts four loss functions on the teacher–student model combination Resnet32×4 vs Resnet8×4, among which the L2 loss function is optimal, as shown in Table 3. For feature matching tasks, L2 Loss directly optimises the element-by-element differences of the feature map, while cross-entropy loss is applicable to classification labels and cannot

**Table 2 Top-1 test accuracy (%) of various knowledge distillation approaches on CIFAR-100.** The bold text indicates the best classification accuracy.

| Student | ShuffleNetV1 | WRN-16-2 | ShuffleNetV2 | MobileNetV2 | MobileNetV2×2 | WRN-40-2 | ShuffleNetV2×1.5 |
|---|---|---|---|---|---|---|---|
| Acc | 71.36 ± 0.25 | 73.51 ± 0.32 | 72.60 ± 0.12 | 64.88 ± 0.29 | 69.06 ± 0.10 | 76.35 ± 0.18 | 74.15 ± 0.22 |
| Teacher | ResNet-32×4 | ResNet-32×4 | ResNet-110×2 | WRN-40-2 | ResNet-32×4 | ResNet-32×4 | ResNet-32×4 |
| Acc | 79.42 | 79.42 | 78.18 | 75.61 | 79.42 | 79.42 | 79.42 |
| KD | 74.30 ± 0.16 | 74.90 ± 0.29 | 76.05 ± 0.34 | 67.58 ± 0.16 | 72.43 ± 0.32 | 77.70 ± 0.13 | 76.82 ± 0.23 |
| FitNet | 74.52 ± 0.03 | 74.70 ± 0.35 | 76.02 ± 0.21 | 67.01 ± 0.20 | 73.09 ± 0.46 | 77.69 ± 0.23 | 77.12 ± 0.24 |
| AT | 75.55 ± 0.19 | 75.38 ± 0.18 | 76.84 ± 0.19 | 67.00 ± 0.29 | 73.08 ± 0.14 | 78.45 ± 0.24 | 77.51 ± 0.31 |
| SP | 74.69 ± 0.32 | 75.16 ± 0.32 | 76.60 ± 0.22 | 66.69 ± 0.19 | 72.99 ± 0.27 | 78.34 ± 0.08 | 77.18 ± 0.19 |
| VID | 74.76 ± 0.22 | 74.85 ± 0.35 | 76.44 ± 0.32 | 66.98 ± 0.23 | 72.70 ± 0.22 | 77.96 ± 0.33 | 77.11 ± 0.35 |
| OFD | 74.86 ± 0.11 | 74.96 ± 0.15 | 76.81 ± 0.22 | 67.06 ± 0.15 | 72.88 ± 0.12 | 78.01 ± 0.13 | 77.31 ± 0.14 |
| CRD | 75.34 ± 0.24 | 75.65 ± 0.08 | 76.67 ± 0.27 | 67.20 ± 0.25 | 73.67 ± 0.26 | 78.15 ± 0.14 | 77.66 ± 0.22 |
| SRRL | 75.18 ± 0.39 | 75.46 ± 0.13 | 76.71 ± 0.27 | 67.33 ± 0.18 | 73.48 ± 0.36 | 78.39 ± 0.19 | 77.55 ± 0.26 |
| SemCKD | 76.31 ± 0.20 | 75.65 ± 0.23 | 77.67 ± 0.30 | 67.57 ± 0.19 | 73.98 ± 0.32 | 78.74 ± 0.17 | 79.13 ± 0.41 |
| ReviewKD | 75.43 ± 0.12 | 75.63 ± 0.16 | 76.87 ± 0.10 | 67.60 ± 0.11 | 73.86 ± 0.12 | 78.22 ± 0.09 | 77.89 ± 0.10 |
| SimKD | 77.18 ± 0.26 | 77.17 ± 0.32 | 78.25 ± 0.24 | 68.19 ± 0.19 | 75.43 ± 0.26 | 79.29 ± 0.11 | 79.54 ± 0.26 |
| DKD | 76.45 ± 0.17 | 75.82 ± 0.15 | 77.79 ± 0.10 | 67.97 ± 0.13 | 74.28 ± 0.14 | 78.80 ± 0.17 | 77.97 ± 0.10 |
| HSAKD | 76.93 ± 0.17 | 77.06 ± 0.10 | 78.16 ± 0.19 | 68.05 ± 0.13 | 75.06 ± 0.17 | 79.34 ± 0.16 | 79.26 ± 0.10 |
| SDD | 76.30 ± 0.16 | 75.75 ± 0.13 | 77.75 ± 0.09 | 67.61 ± 0.21 | 74.11 ± 0.16 | 78.79 ± 0.19 | 79.15 ± 0.21 |
| MFAKD | **78.38 ± 0.13** | **78.88 ± 0.24** | **78.28 ± 0.16** | **69.22 ± 0.31** | **75.98 ± 0.10** | **79.81 ± 0.23** | **80.04 ± 0.15** |

**Table 3 Accuracies of teacher–student model combinations of ResNet32×4 and ResNet8×4 using different loss functions on the CIFAR-100 dataset.**

| Loss function | L2 loss | Smooth L2 loss | Cross entropy loss | RMSE loss |
|---|---|---|---|---|
| Accuracy | 79.90 ± 0.08 | 79.25 ± 0.11 | 79.63 ± 0.15 | 79.75 ± 0.15 |

directly constrain the similarity of continuous features. The accuracy standard deviation (±0.08) of L2 in the data is the smallest, indicating that its training stability is higher. This is because L2 uniformly weights all errors, avoiding the uncertainty caused by the segmented gradient of Smooth L2 or the scale variation of RMSE. Especially in feature distillation, stable gradients help optimise the consistency of high-dimensional features.

As shown in Table 4, when using four randomly selected teacher–student model combinations on the TinyImagNet dataset, the MFAKD method outperforms other mainstream knowledge distillation methods in terms of Top-1 accuracy.

## Comparison of loss functions

In addition, the experiment did not set traditional classification losses, possibly because when these two losses coexist, the gradient update direction is not consistent. As shown in

Table 5, as the traditional classification loss gradually increases, the performance of the student model gradually decreases.

## Ablation experiments

Multilevel feature alignment knowledge distillation is the process of aligning the student features to the teacher features in both the channel and spatial dimensions. Therefore, this section analyses the ablation experiment for spatial dimension alignment and channel alignment. Table 6 shows the experiments using MBCA and spatial alignment separately.

### Spatial dimension alignment ablation experiment

Spatial dimension alignment allows the students to learn more comprehensive teacher knowledge. *Gholamalinezhad & Khosravi (2020)* believes that pooling operations reduce the size of feature maps, thereby reducing the computational complexity and memory consumption of the model. Pooling operations have a disadvantage: they average or maximize pixel values in local areas, which blurs detailed features and may cause the teacher's fine-grained information to be lost. This inaccuracy in handling image details arises from reducing feature map size to lower computational complexity and memory consumption. To enable the student model to learn comprehensive knowledge from the teacher, we align student features with the teacher's feature map in spatial dimensions *via* upsampling. This operation matches the teacher's feature size, ensuring more effective information alignment between the two models. After the experiment, the student network can obtain richer and more detailed information from the teacher network, helping to improve the distillation effect. The experimental results are shown in the MFAKD in Tables 7 and 8, where "-" indicates that the spatial dimension alignment is not needed. Two teacher–student model combinations (teacher WRN-40-2 with student MobileNetV2 and teacher ResNet-32×4 with student MobileNetV2×2) have a spatial dimension of $8 \times 8$ for the teacher features and $2 \times 2$ for the student features. The remaining groups of teacher–student model combinations have a spatial dimension of $8 \times 8$ for the teacher features, and the spatial dimension of the student features is $4 \times 4$. The experimental results indicate that smaller student features aligned spatially with teacher features can lead to better distillation effects.

### Channel alignment ablation experiment

This subsection separately analyzes the effect of each branch and the multiple branches as projectors on the distillation effect. As shown in Tables 7 and 8, the sandglass projector outperforms the bottleneck projector in terms of test accuracy for all 14 pairs of student-teacher model combinations. For some teacher–student model combinations, the test accuracy of the sandglass+ projector is slightly lower than that of the bottleneck module. Examples of such teacher–student model combinations include three groups: ShuffleNetV2 and ResNet-32×4, ShuffleNetV2×1.5 and ResNet-32×4, and MobileNetV2×2 and ResNet-32×4. The core of the Shufflenet network architecture lies in group convolution and channel shuffle, while ResNet-32×4 employs ordinary convolution and residual connections; the core of MobileNet is the inverted residual block, whereas ResNet-32×4 uses the Bottleneck Block. These differences may be the reason why deeper

**Table 4 Top-1 test accuracy (%) of various knowledge distillation approaches on tinyImageNet.** The bold text indicates the best classification accuracy.

| Student | Vgg-8 | Vgg-11 | ResNet-10 | ResNet-10 |
|---|---|---|---|---|
| Acc | 54.61 ± 0.13 | 58.60 ± 0.11 | 58.01 ± 0.14 | 58.01 ± 0.19 |
| Teacher | Vgg-19 | Vgg-16 | ResNet-34 | ResNet-50 |
| Acc | 61.15 | 61.22 | 65.61 | 64.98 |
| KD | 55.55 ± 0.11 | 61.51 ± 0.13 | 58.92 ± 0.08 | 58.63 ± 0.17 |
| FitNet | 55.24 ± 0.16 | 59.08 ± 0.09 | 58.22 ± 0.18 | 57.76 ± 0.14 |
| AT | 53.55 ± 0.18 | 61.40 ± 0.11 | 59.16 ± 0.13 | 58.92 ± 0.19 |
| SP | 55.09 ± 0.11 | 61.61 ± 0.14 | 55.91 ± 0.21 | 57.17 ± 0.18 |
| VID | 54.95 ± 0.11 | 60.07 ± 0.11 | 58.53 ± 0.13 | 57.65 ± 0.17 |
| CRD | 56.99 ± 0.10 | 62.04 ± 0.11 | 60.02 ± 0.17 | 59.31 ± 0.13 |
| ReviewKD | 57.01 ± 0.17 | 62.24 ± 0.18 | 60.32 ± 0.15 | 59.43 ± 0.23 |
| SimKD | 59.84 ± 0.13 | 61.55 ± 0.11 | 63.94 ± 0.14 | 64.87 ± 0.21 |
| HSAKD | 59.60 ± 0.10 | 61.25 ± 0.19 | 62.89 ± 0.19 | 64.66 ± 0.11 |
| SDD | 58.01 ± 0.18 | 61.01 ± 0.21 | 61.82 ± 0.19 | 60.87 ± 0.21 |
| MFAKD | **62.17 ± 0.09** | **62.05 ± 0.11** | **66.10 ± 0.13** | **66.11 ± 0.10** |

**Table 5 Test the accuracy of ResNet32×4 and ResNet8×4 in the Top1 of CIFAR100.** Among them, $\alpha$ represents the loss of the teacher model and the student model, while $\beta$ represents the classification loss.

| $\alpha$ | $\beta$ | Accuracy |
|---|---|---|
| 1.0 | 0.0 | 79.90 ± 0.08 |
| 1.0 | 0.2 | 79.42 ± 0.13 |
| 1.0 | 0.4 | 79.17 ± 0.11 |
| 1.0 | 0.6 | 79.02 ± 0.17 |

**Table 6 Ablations that isolate the spatial dimension and MBCA modules, Among them, SDA represents spatial alignment.** The bold text indicates the best classification accuracy.

| Student | ShuffleNetV1 | ShuffleNetV2 | MobileNetV2 | MobileNetV2×2 | ShuffleNetV2×1.5 |
|---|---|---|---|---|---|
| Acc | 71.36 ± 0.25 | 72.60 ± 0.12 | 64.88 ± 0.29 | 69.06 ± 0.10 | 74.15 ± 0.22 |
| Teacher | ResNet-32×4 | ResNet-110×2 | WRN-40-2 | ResNet-32×4 | ResNet-32×4 |
| Acc | 79.42 | 78.18 | 75.61 | 79.42 | 79.42 |
| SDA | 74.22 ± 0.16 | 75.45 ± 0.18 | 64.13 ± 0.18 | 71.88 ± 0.10 | 75.18 ± 0.20 |
| MBCA | 77.91 ± 0.15 | 77.88 ± 0.18 | 67.93 ± 0.10 | 75.01 ± 0.11 | 79.48 ± 0.12 |
| MBCA+SDA | **78.38 ± 0.13** | **78.29 ± 0.16** | **69.32 ± 0.11** | **75.98 ± 0.10** | **80.04 ± 0.15** |

feature projectors perform poorly. In this study, by concatenating multiple weak projector branches into a single strong projector, MBCA, the student model performance is further improved, as shown by the MBCA results in Tables 7 and 8. To save space, we chose to draw line charts using the data from Table 7, as shown in Figs. 4, 5, and 6.

**Table 7 Test accuracies using different projectors on the CIFAR-100 dataset.** The bold text indicates the best classification accuracy.

| Student | WRN-40-1 | ResNet-8×4 | ResNet-110 | ResNet-116 | VGG-8 | ResNet-8×4 | ShuffleNetV2 |
|---|---|---|---|---|---|---|---|
| Acc | 71.92 ± 0.17 | 73.09 ± 0.30 | 74.37 ± 0.17 | 74.46 ± 0.09 | 70.46 ± 0.29 | 73.09 ± 0.30 | 72.60 ± 0.12 |
| Flops | 40.60 M | 92.1 M | 114.9 M | 121.1 M | 201.1 M | 92.1 M | 43.2 M |
| Params | 0.57 M | 1.23 M | 1.73 M | 1.83 M | 3.96 M | 1.23 M | 1.35 M |
| Infer time | 8.2 ms | 0.7 ms | 6.7 ms | 9.3 ms | 1.3 ms | 0.7 ms | 11.5 ms |
| Teacher | WRN-40-2 | ResNet-32×4 | ResNet-110×2 | ResNet-110×2 | ResNet-32×4 | WRN-40-2 | ResNet-32×4 |
| Acc | 75.61 | 79.42 | 78.18 | 78.18 | 79.42 | 75.61 | 79.42 |
| Flops | 156.8 M | 527.5 M | 454.1 M | 454.1 M | 527.5 M | 156.8 M | 527.5 M |
| Params | 2.25 M | 7.43 M | 6.91 M | 6.91 M | 7.43 M | 2.25 M | 7.43 M |
| Infer time | 9.8 ms | 2.7 ms | 9.1 ms | 9.1 ms | 2.7 ms | 9.8 ms | 2.7 ms |
| Bottleneck | 73.13 ± 0.17 | 78.08 ± 0.15 | 77.82 ± 0.15 | 77.90 ± 0.11 | 75.76 ± 0.18 | 76.75 ± 0.23 | 78.39 ± 0.27 |
| Δ Flops | 3.21 M | 13.76 M | 3.21 M | 3.21 M | 3.96 M | 3.99 M | 3.86 M |
| Δ Params | 0.04 M | 0.21 M | 0.04 M | 0.04 M | 0.24 M | 0.06 M | 0.24 M |
| Sandglass | 74.54 ± 0.19 | 79.33 ± 0.20 | 77.97 ± 0.17 | 77.99 ± 0.12 | 75.78 ± 0.21 | 76.80 ± 0.26 | 78.26 ± 0.21 |
| Δ Flops | 15.31 M | 79.92 M | 15.31 M | 15.31 M | 29.41 M | 19.47 M | 27.64 M |
| Δ Params | 0.13 M | 0.64 M | 0.13 M | 0.23 M | 0.93 M | 0.25 M | 0.72 M |
| Sandglass+ | 75.01 ± 0.20 | 79.63 ± 0.11 | 77.97 ± 0.19 | 78.07 ± 0.10 | 75.91 ± 0.23 | 76.79 ± 0.21 | 78.28 ± 0.27 |
| Δ Flops | 25.88 M | 122.0M | 25.88 M | 25.88 M | 39.94 M | 30.04 M | 38.17 M |
| Δ Params | 0.20 M | 1.01 M | 0.20 M | 0.40 M | 1.49 M | 0.32 M | 1.38 M |
| MBCA | 75.62 ± 0.16 | 79.90 ± 0.08 | 78.08 ± 0.11 | 78.19 ± 0.11 | 76.10 ± 0.17 | 77.39 ± 0.18 | 78.50 ± 0.15 |
| Δ Flops | 44.42 M | 215.7 M | 44.42 M | 44.42 M | 293.3 M | 53.50 M | 69.67 M |
| Δ Params | 0.39 M | 1.65 M | 0.39 M | 0.69 M | 2.57 M | 0.64 M | 2.34 M |
| MFAKD | - | - | - | - | **76.27 ± 0.13** | - | **78.60 ± 0.14** |

## Analytical experiments

This section uses different upsampling methods for distillation performance analysis in spatial dimension alignment, and analyzes MBCA using the center kernel align (CKA) and the expected calibration error (ECE) evaluation indicators in channel alignment.

### Upsampling analysis experiment

In order to explore the impact of different upsampling methods on MFAKD, the MFAKD method attempted common nearest neighbor interpolation, bilinear interpolation, and transpose convolution methods using three sets of teacher–student model combinations. As shown in Table 9, the nearest neighbor interpolation performs well, which may be attributed to its ability to accurately maintain the position and shape of boundaries.

### Multibranch channel alignment analysis experiment

In order to analyse the impact of projectors on distillation performance, this section uses the radial basis function CKA similarity proposed by *Kornblith et al. (2019)* to measure the

**Table 8 Test accuracies and computational costs using different projectors on the CIFAR-100 dataset.** The bold text indicates the best classification accuracy.

| Student | ShuffleNetV1 | WRN-16-2 | ShuffleNetV2 | MobileNetV2 | MobileNetV2×2 | WRN-40-2 | ShuffleNetV2×1.5 |
|---|---|---|---|---|---|---|---|
| Acc | 71.36 ± 0.25 | 73.51 ± 0.32 | 72.60 ± 0.12 | 64.88 ± 0.29 | 69.06 ± 0.10 | 76.35 ± 0.18 | 74.15 ± 0.22 |
| Flops | 229.6 M | 57.80 M | 43.20 M | 111.9 M | 444.3 M | 156.8 M | 67.53 M |
| Params | 0.94 M | 0.70 M | 1.35 M | 0.81 M | 2.35 M | 2.25 M | 2.58 M |
| Infer time | 12.8 ms | 4.3 ms | 11.5 ms | 9.06 ms | 10.8 ms | 9.8 ms | 13.3 ms |
| Teacher | ResNet-32×4 | ResNet-32×4 | ResNet-110×2 | WRN-40-2 | ResNet-32×4 | ResNet-32×4 | ResNet-32×4 |
| Acc | 79.42 | 79.42 | 78.18 | 75.61 | 79.42 | 79.42 | 79.42 |
| Flops | 527.5 M | 527.5 M | 454.1 M | 156.8 M | 527.5 M | 527.5 M | 527.5 M |
| Params | 7.43 M | 7.43 M | 6.91 M | 2.25 M | 7.43 M | 7.43 M | 7.43 M |
| Infer time | 2.7 ms | 2.7 ms | 9.1 ms | 9.06 ms | 9.8 ms | 2.7 ms | 2.7 ms |
| Bottleneck | 77.18 ± 0.26 | 77.17 ± 0.32 | 78.25 ± 0.24 | 67.69 ± 0.19 | 75.43 ± 0.26 | 79.29 ± 0.11 | 79.54 ± 0.26 |
| Δ Flops | 4.88 M | 12.71 M | 1.21 M | 0.22 M | 0.89 M | 12.71 M | 4.35 M |
| Δ Params | 0.30 M | 0.19 M | 0.07 M | 0.05 M | 0.22 M | 0.19 M | 0.27 M |
| Sandglass | 77.31 ± 0.20 | 78.11 ± 0.27 | 78.26 ± 0.19 | 69.26 ± 0.20 | 75.32 ± 0.19 | 79.85 ± 0.19 | 79.68 ± 0.26 |
| Δ Flops | 45.93 M | 61.04 M | 11.20 M | 1.39 M | 5.58 M | 61.04 M | 36.49 M |
| Δ Params | 1.46 M | 0.55 M | 0.69 M | 0.18 M | 0.79 M | 0.45 M | 1.27 M |
| Sandglass+ | 77.45 ± 0.22 | 78.76 ± 0.24 | 78.28 ± 0.20 | 67.69 ± 0.17 | 75.00 ± 0.29 | 79.82 ± 0.17 | 79.06 ± 0.11 |
| Δ Flops | 56.45 M | 103.1 M | 13.84 M | 2.06 M | 8.21 M | 103.1 M | 47.02 M |
| Δ Params | 1.72 M | 1.10 M | 0.86 M | 0.41 M | 1.35 M | 1.10 M | 2.03 M |
| MBCA | 78.31 ± 0.19 | 78.88 ± 0.25 | 78.25 ± 0.17 | 68.03 ± 0.18 | 75.41 ± 0.18 | 79.81 ± 0.13 | 79.58 ± 0.27 |
| Δ Flops | 429.1 M | 176.93 M | 105.0 M | 58.97 M | 235.1 M | 176.9 M | 351.5 M |
| Δ Params | 3.48 M | 1.84 M | 1.63 M | 0.64 M | 2.36 M | 1.74 M | 3.57 M |
| MFAKD | **78.38 ± 0.13** | - | **78.29 ± 0.16** | **69.32 ± 0.11** | **75.98 ± 0.10** | - | **80.04 ± 0.15** |

similarity between teacher and student features in different projectors, and uses the expected calibration error (ECE) proposed by *Guo et al. (2017)* to evaluate the performance of each branch and multiple branches. Finally, analyse the differences in teacher–student feature loss between the training and testing sets.

**Multibranch channel alignment can combine teacher–student model similarity.** A higher teacher–student model CKA similarity indicates that the student features are closer to the teacher features, as described in *Kornblith et al. (2019)* and *Chen et al. (2023)*. Specifically, CKA takes two feature representations X and Y as inputs and calculates their normalized similarity based on the Hilbert Schmidt Information (HSIC):

$$HSIC(K, L) = \frac{1}{n(n-3)} \left( \text{tr}(\tilde{K}\tilde{L}) + \frac{1^\top \tilde{K} 11^\top \tilde{L} 1}{(n-1)(n-2)} - \frac{2}{n-2} 1^\top \tilde{K}\tilde{L} 1 \right). \tag{6}$$

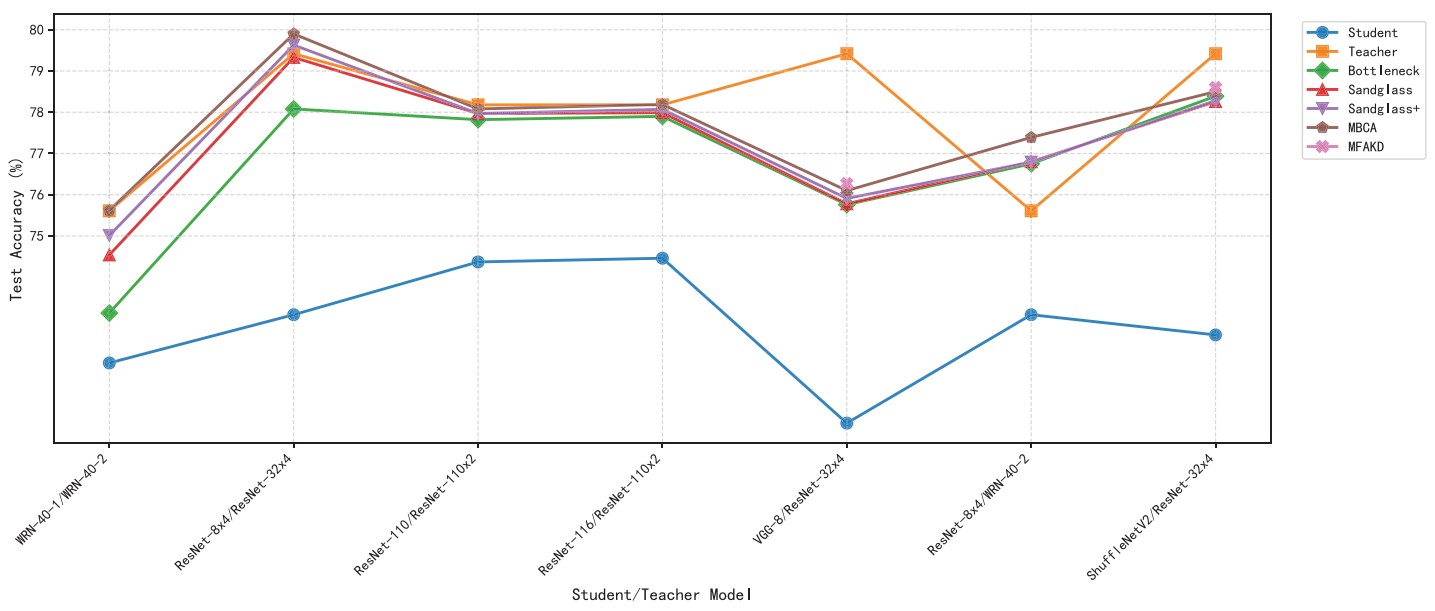

**Figure 4 Test accuracies on CIFAR-100 dataset.**

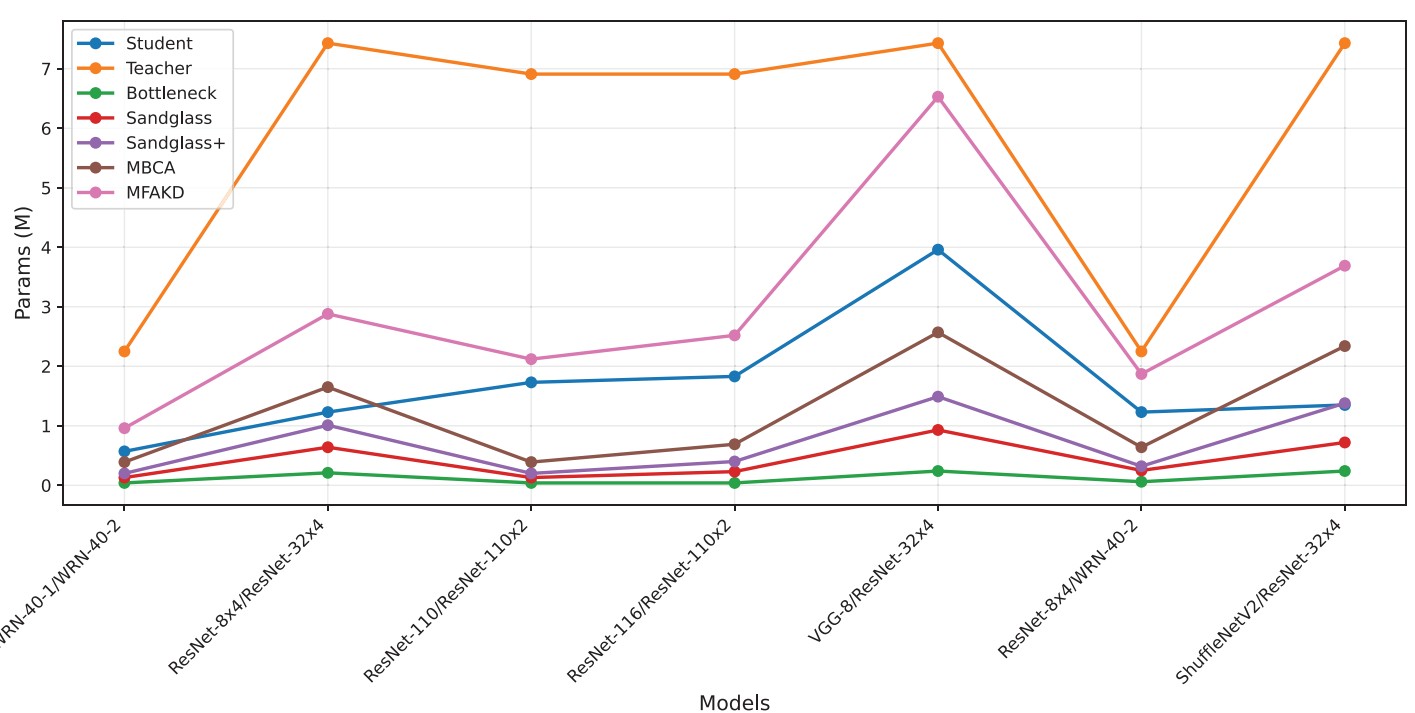

**Figure 5 Flops for various teacher–student combinations and projectors on CIFAR-100.**

Among them, $\tilde{K}$ and $\tilde{L}$ are obtained by setting the diagonal entries of the similarity matrices K and L to zero. Then, CKA similarity can be calculated by averaging the HISC scores across k small batches.
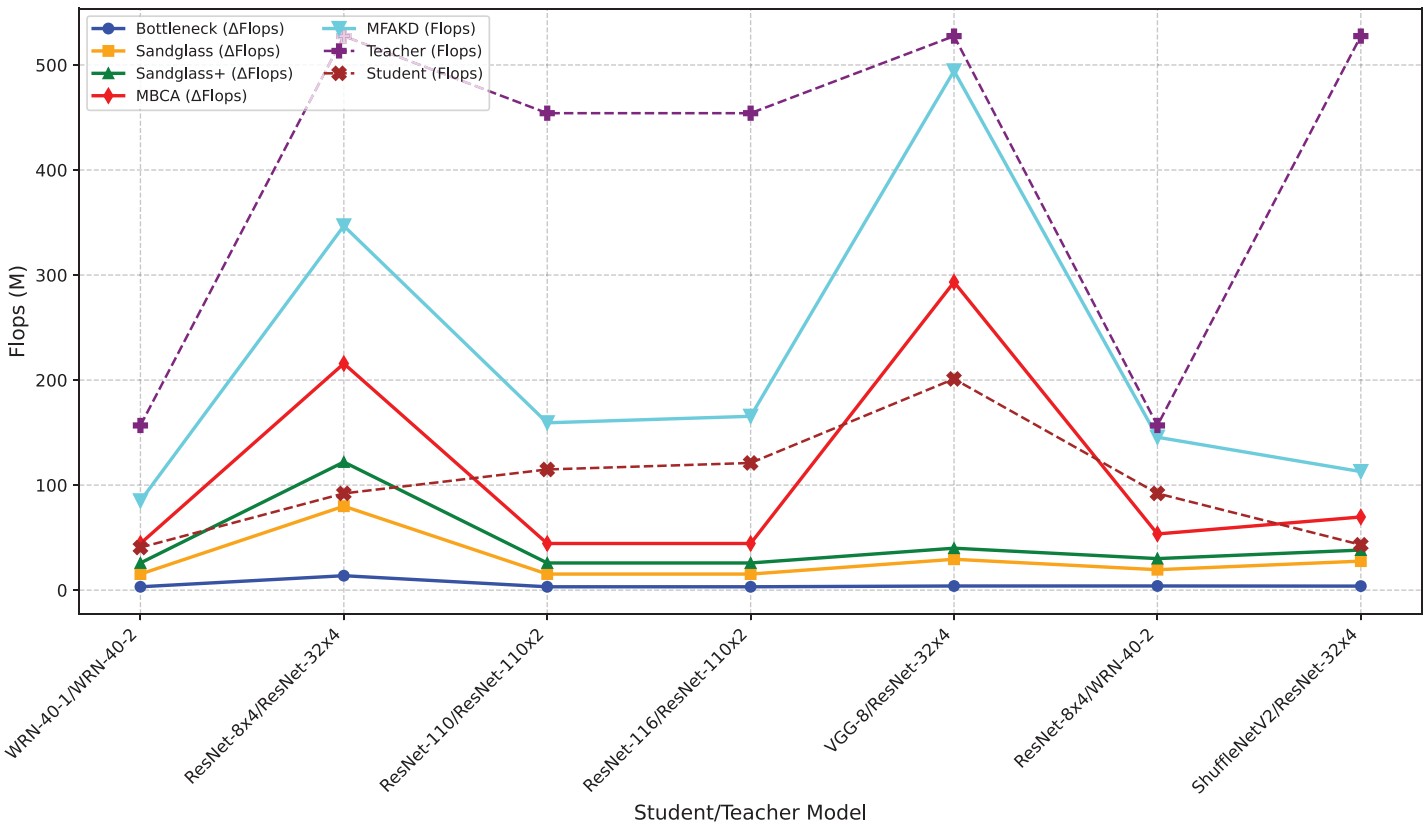

**Figure 6** Params for various teacher–student combinations and projectors on CIFAR-100.

**Table 9 Top-1 test accuracy (%) of various knowledge distillation approaches on the CIFAR-100 dataset.**

| Student | ShuffleNetV2×1.5 | MobileNetV2 | VGG-8 |
|---|---|---|---|
| Acc | 74.15 ± 0.22 | 65.43 ± 0.29 | 70.46 ± 0.29 |
| Params | (2.58 M) | (0.81 M) | (3.96 M) |
| Teacher | ResNet32×4 | WRN-40-2 | ResNet32×4 |
| Acc | 79.42 | 75.61 | 79.42 |
| Params | (7.43 M) | (2.25 M) | (7.43 M) |
| Nearest neighbor interpolation | 80.04 ± 0.15 | 69.22 ± 0.11 | 76.27 ± 0.13 |
| Bilinear interpolation | 79.55 ± 0.11 | 68.63 ± 0.15 | 76.07 ± 0.19 |
| Transposed convolution | 79.73 ± 0.18 | 68.93 ± 0.15 | 76.17 ± 0.19 |

$$CKA = \frac{\frac{1}{k}\sum_1^k HSIC(X_iX_i^\top, Y_iY_i^\top)}{\sqrt{\frac{1}{k}\sum_1^k HSIC(X_iX_i^\top, X_iX_i^\top)}\sqrt{\frac{1}{k}\sum_1^k HSIC(Y_iY_i^\top, Y_iY_i^\top)}}. \tag{7}$$

Among them, $X_i \in \mathbb{R}^{B \times p1}$ and $Y_i \in \mathbb{R}^{B \times p2}$ are matrices containing the activation values of the i-th small batch B samples.

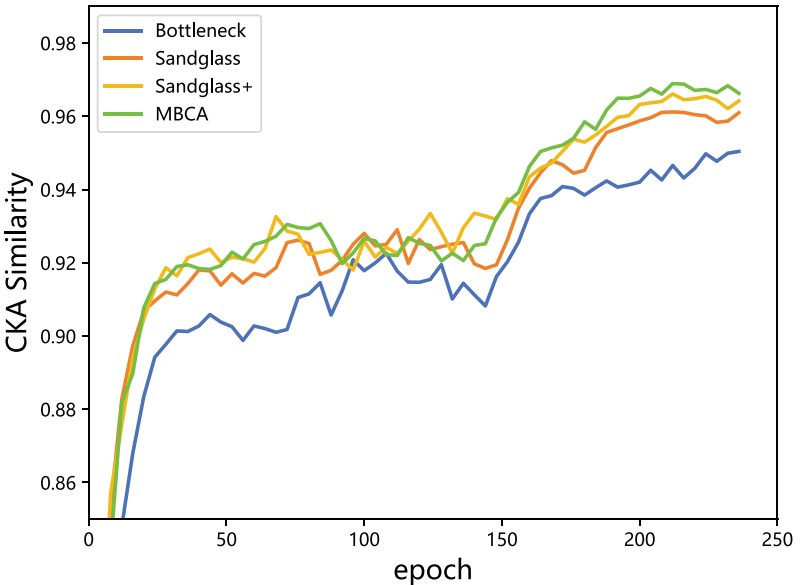

**Figure 7** CKA similarity between the teacher model ResNet32×4 and the student model ResNet8×4 on the training set of the CIFAR-100 dataset.

This subsection monitors the radial basis function (RBF) CKA similarity between the student features and the teacher features through feature distillation. With the combination of the Resnet32×4 teacher and the Resnet8×4 student, the experiment is performed using different projectors (different branches) for the teacher–student model CKA similarity calculation, as shown in Fig. 7. From the results, a greater the CKA similarity for the teacher–student features correlates to a better distillation and indicates that the use of projectors reduces the knowledge gap between the teacher model and the student model and leads to more appropriate the teacher model knowledge being delivered to the the student model. Adding a strong projector to students can increase the teacher–student model similarity and is an effective method for improving the classification performance.

**Multibranch channel alignment can mitigate teacher overconfidence.** The confidence is the degree of confidence or trustworthiness of a model in its predictions. In classification tasks, confidence enables us to understand how reliable the model's predictions for specific categories are, which helps evaluate the performance of the model and determine the accuracy of the predictions. Confidence calibration can help models more accurately evaluate the reliability of their predictions. *Guo et al. (2017)* believes that through calibration, the confidence level of the model can more accurately reflect the probability of actual events occurring, thereby improving the predictive accuracy of the model. For example, *Jiang et al. (2012)* believes that in automated medical diagnosis, when the confidence level of the disease diagnosis network is low, control needs to be handed over to human doctors. This subsection investigates the effect of each single-branch projector as

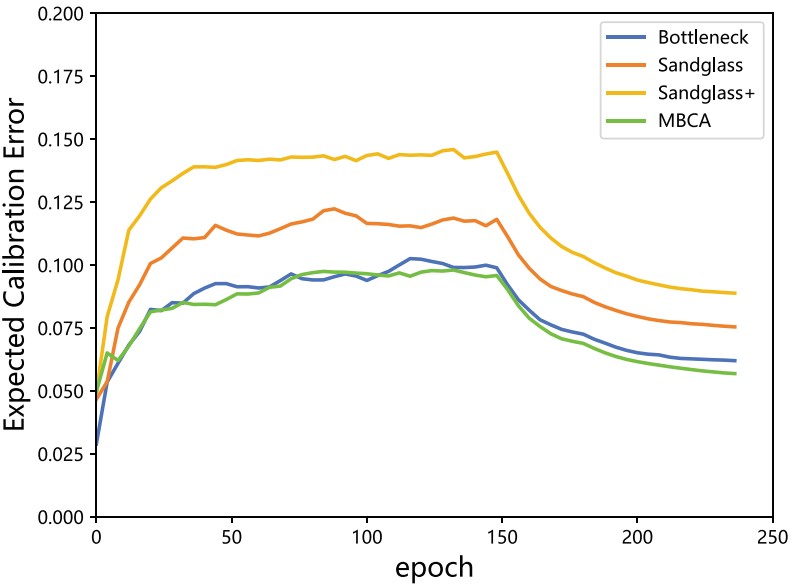

**Figure 8 Expected calibration error of the teacher model ResNet32×4 *vs* the student model ResNet8×4 on the test set of the CIFAR-100 dataset.**

well as the multibranch projector on the calibration properties of the student model. As mentioned in existing work (*Guo et al., 2017*), increasing the size of the model tends to result in a lower calibration. For this reason, this subsection averages the expected calibration error of the student model projections into T regions and calculates the expected calibration error values as shown in Eq. (8):

$$M_{ECE} = \sum_{i=1}^{T} \frac{|B_i|}{n_{test}} |M_{ACC}(B_i) - M_{CON}(B_i)| \tag{8}$$

where $n_{test}$ denotes the number of test samples and $B_i$ is the indexed set of test samples with the largest prediction score in the $i$th region.

$$M_{ACC}(B_i) = \frac{1}{|(B_i)|} \sum_{j \in B_i} 1(y_{pre}^j = y^j) \tag{9}$$

where $y_{pre}^j$ is the predicted category and $y^j$ is the true category of the $j$th sample.

$$M_{CON}(B_i) = \frac{1}{|B_i|} \sum_{j \in B_i} h^j \tag{10}$$

where $h^j$ is the highest prediction score for the $j$th sample.

The $M_{ECE}$ is used to measure the gap between the accuracy and confidence; thus, a lower $M_{ECE}$ score is better. In this subsection, the $M_{ECE}$ values of the expected calibration error of the teacher model and the student model in different projectors (different branches) are calculated and plotted in a line graph, as shown in Fig. 8.

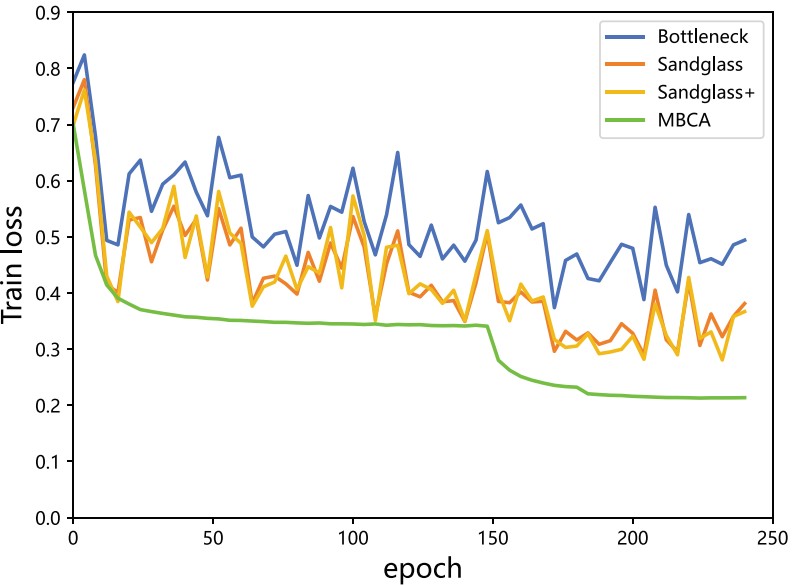

**Figure 9** Training loss of the teacher model ResNet32×4 *vs* the student model ResNet8×4 on the training set of the CIFAR-100 dataset.

According to the results in Fig. 8, it can be seen that although using Sandglass and Sandglass+ Projector will achieve better distillation results than Bottleneck Projector, they will make the model overconfident. This study finds that the design of multibranch projectors can alleviate the overconfidence of the model by parallelizing three different projectors. This study speculates that the student model does not need to overly imitate the teacher model, but rather softens the teacher's information from the perspective of confidence calibration.

**Multibranch channel alignment can make the optimization direction of the model more stable.** This section uses different feature projectors on the training and testing sets to monitor the changes in feature loss values for the teacher model and the student model. As shown in Figs. 9 and 10, using multibranch channel alignment on the training and testing sets yields better results, indicating that student features are closer to teacher features. It is worth noting that the alignment effect of multiple branch channels makes the optimization direction smoother and more stable, while using a single branch for optimization results in a steeper direction.

## DISCUSSION

The first problem is that the study in this paper increases the number of parameters. For example, in the teacher–student model combination with the best distillation effect, for the teacher WRN-40-2 and the student ResNet-8×4, the total number of senators for the teacher is approximately 2.255 M, and the number of senators for the student is approximately 1.233 M. The method in this paper adds approximately 0.6 M additional

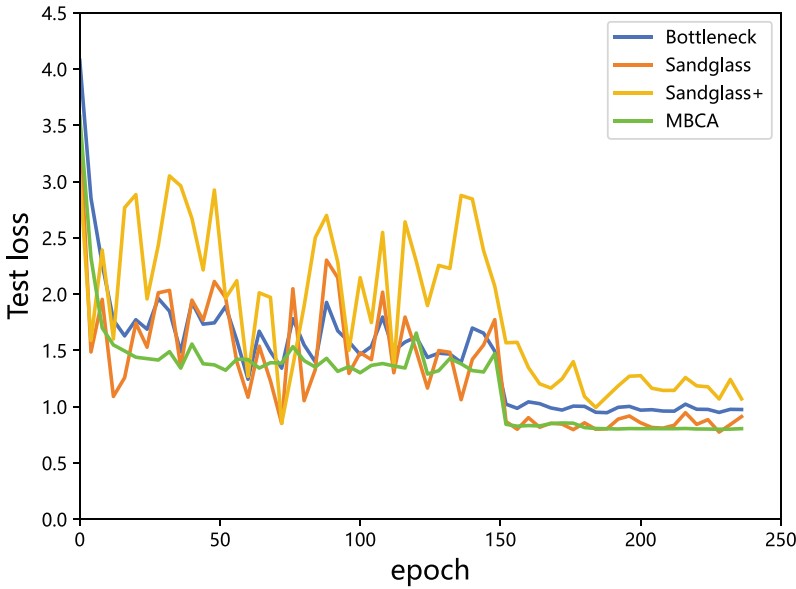

**Figure 10 Testing loss of the teacher model ResNet32×4 *vs* the student model ResNet8×4 on the testing set of the CIFAR-100 dataset.**

parameters. Therefore, future research is needed on how to design lighter and more efficient projectors.

# FUTURE WORK

Future work can focus on designing more lightweight and efficient projectors, such as reducing extra parameters *via* lightweight architectures, parameter sharing, or parameter-free projectors methods. Meanwhile, exploring combinations with specific optimization approaches like model pruning, quantization, low-rank decomposition, and neural architecture search (NAS) will help balance model performance and parameter count to enhance overall distillation efficiency.

# CONCLUSIONS

In this study, we explore the impact of teacher–student networks on the spatial alignment *vs* channel alignment; here, Bottleneck, Sandglass, Sandglass+, and MBCA projectors are used to allow the student feature maps to be aligned with the teacher feature maps from the channel and from the spatial dimensions. We design experiments to validate the superiority of this method, obtain state-of-the-art results, and perform a detailed analysis of the student-teacher network using the expected calibration error and center kernel alignment; our experimental results demonstrate that multibranch projectors can produce more compatible student features with the teacher features and mitigate the overconfidence of the teacher network. We hope that this study can contribute to future research on feature alignment between the teacher model and the student model.

### Funding

This study was funded by the China Chongqing Municipal Science and Technology Bureau, grant number CSTB2024TIAD-CYKJCXX0009, CSTB2024NSCQ-LZX0043; Chongqing Municipal Commission of Housing and Urban-Rural Development, grant number CKZ2024-87; the Chongqing University of Technology graduate education high-quality development project, grant number gzlsz202401; the Chongqing University of Technology-Chongqing LINGLUE Technology Co., Ltd.. Electronic Information (artificial intelligence) graduate joint training base; the Postgraduate Education and Teaching Reform Research Project in Chongqing, grant number yjg213116; and the Chongqing University of Technology-CISDI Chongqing Information Technology Co., LTD. Computer Technology graduate joint training base. The funders had no role in study design, data collection and analysis, decision to publish, or preparation of the manuscript.

### Grant Disclosures

The following grant information was disclosed by the authors:
China Chongqing Municipal Science and Technology Bureau: CSTB2024TIAD-CYKJCXX0009, CSTB2024NSCQ-LZX0043.
Chongqing Municipal Commission of Housing and Urban-Rural Development: CKZ2024-87.
Chongqing University of Technology Graduate Education High-Quality Development Project: gzlsz202401.
Chongqing University of Technology-Chongqing LINGLUE Technology Co., Ltd. Electronic Information (artificial intelligence) Graduate Joint Training Base.
Postgraduate Education and Teaching Reform Research Project in Chongqing: yjg213116.
Chongqing University of Technology-CISDI Chongqing Information Technology Co., LTD.
Computer Technology Graduate Joint Training Base.

### Competing Interests

The authors declare that they have no competing interests.

### Author Contributions

- Yang Zhang conceived and designed the experiments, performed the experiments, authored or reviewed drafts of the article, and approved the final draft.
- Pan He conceived and designed the experiments, performed the experiments, analyzed the data, performed the computation work, prepared figures and/or tables, authored or reviewed drafts of the article, and approved the final draft.
- Chuanyun Xu conceived and designed the experiments, authored or reviewed drafts of the article, and approved the final draft.
- Jingyan Pang conceived and designed the experiments, performed the experiments, authored or reviewed drafts of the article, and approved the final draft.

- Xiao Wang conceived and designed the experiments, performed the experiments, prepared figures and/or tables, authored or reviewed drafts of the article, and approved the final draft.
- Xinghai Yuan analyzed the data, performed the computation work, authored or reviewed drafts of the article, and approved the final draft.
- Pengfei Lv analyzed the data, performed the computation work, authored or reviewed drafts of the article, and approved the final draft.
- Gang Li conceived and designed the experiments, analyzed the data, authored or reviewed drafts of the article, and approved the final draft.

### Data Availability

The code is available at GitHub:

- https://github.com/abcdpan/mfakd.

- Pan, H. (2025). Aligning to the teacher: multilevel feature-aligned knowledge distillation. Zenodo. https://doi.org/10.5281/zenodo.15851926.

The cifar-100 dataset: https://www.cs.toronto.edu/~kriz/cifar.html.

The tiny-ImageNet dataset: http://cs231n.stanford.edu/tiny-imagenet-200.zip.

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
