# Peer review of "Aligning to the teacher: multilevel feature-aligned knowledge distillation"

_PeerJ Computer Science, doi:10.7717/peerj-cs.3075_

## Round 0.1 · original submission · Major Revisions

Thank you for submitting your manuscript. After weighing the reviewers’ detailed critiques, I must request a major revision. While the multilevel feature-alignment idea is promising, the current manuscript lacks sufficient empirical depth and presentation clarity to support its claims.

Please expand the experimental scope: benchmark MFAKD against more recent distillation methods (e.g., Scaled Decoupled Distillation, CRD, DKD, OFD, ReviewKD, HSAKD), add ablations that isolate the spatial-alignment and MBCA modules, and report computational overhead (parameters, FLOPs, training/inference time) alongside statistical significance tests for accuracy gains. Address formatting and language issues, consolidate redundant figures, correct table alignment and citation styles, and split overly long sentences for readability.

Kindly revise accordingly and supply a point-by-point response to each reviewer's comment. We look forward to evaluating your improved submission.

**Language Note:** The review process has identified that the English language must be improved. PeerJ can provide language editing services - please contact us at [email protected] for pricing (be sure to provide your manuscript number and title). Alternatively, you should make your own arrangements to improve the language quality and provide details in your response letter. – PeerJ Staff

Reviewer 1 ·

Basic reporting

This paper introduces Multilevel Feature Alignment Knowledge Distillation (MFAKD), a method to improve knowledge transfer from a large teacher model to a smaller student model. MFAKD aligns features at both spatial and channel levels: a spatial dimension alignment module uses upsampling to match the spatial dimensions of student and teacher feature maps; a multibranch channel alignment (MBCA) module transforms student features to resemble teacher features at the channel level.

By leveraging these alignment techniques and the teacher’s discriminative classifiers, the student model can effectively learn from the teacher and potentially outperform it. Experimental results show the effectiveness of the proposed method.

Experimental design

The experimental results are not sufficient. The compared latest method is SimKD, published at CVPR-2022. More advanced method published in 2024 needs to be compared, such as Scaled decoupled distillation [1].

The previous ReviewKD also uses Multilevel Feature Alignment for distillation. This paper should discuss the related method and compare it.

Another effective way for feature distillation is self-supervised knowledge, such as HSAKD [3]. This paper should discuss the related method and compare it.

Validity of the findings

-

Cite this review as

·

Basic reporting

Language and Clarity:
The manuscript is generally well-written in professional English, but minor grammatical errors and redundant expressions exist (e.g., inconsistent punctuation, overly complex sentence structures). A thorough proofreading is recommended. For example:

In the abstract, "students can utilize their strengths and surpass teachers" should clarify that "students" refers to the student model (e.g., "the student model can leverage its advantages to surpass the teacher model").

Some lengthy sentences (e.g., the analogy to traditional Chinese education in the Introduction) could be split for readability.

Figures and Tables:
Figures 2 and 3 appear redundant (both describe spatial alignment operations); verify and consolidate.

Column alignment in Tables 2 and 3 needs improvement for clarity (e.g., headers and data misaligned).

Citations:
Inconsistent citation formats (e.g., "Chen et al. (2021a)" vs. "Chen et al. (2022)"). Ensure uniformity.

Experimental design

Reproducibility:
Clarify ambiguous terms in training details, e.g., "total training period was set to 240" (specify units, e.g., epochs).

Justify the selection of teacher-student combinations (e.g., why ResNet and WRN architectures were chosen).

Loss Function Analysis:
Table 1 shows that L2 loss performs best, but the rationale for the poor performance of alternatives (e.g., Smooth L2) is missing. Add a brief discussion on loss function selection.

Statistical Significance:
Some improvements are marginal (e.g., MFAKD outperforms SimKD by only 0.5% in Table 2). Include statistical significance tests (e.g., p-values).

Validity of the findings

Ablation Study Enhancements:
In the spatial alignment experiment, "upsampling outperforms pooling" is claimed, but computational costs or parameter changes are not compared. Add an analysis of efficiency trade-offs.

For channel alignment, explain performance fluctuations in Sandglass+ (e.g., lower accuracy than Bottleneck in Table 6).

Parameter Efficiency:
The Discussion notes a 0.6M parameter increase but lacks lightweight design proposals. Suggest future directions (e.g., dynamic projector design).

Additional comments

Figure/Table Optimization:
Add legends to Figures 5 and 6 (e.g., color coding for different projectors).

Expand abbreviations in Table 8 (e.g., "Nearest" → "Nearest Neighbor Interpolation").

Terminology Consistency:
Unify terms (e.g., "multibranch" vs. "multi-branch") throughout the manuscript.

Code and Data:
The GitHub link is provided, but include code documentation for critical hyperparameters (e.g., learning rate decay strategy).

Reviewer 3 ·

Basic reporting

While the proposed Multilevel Feature Alignment Knowledge Distillation (MFAKD) method introduces an innovative approach to improving knowledge distillation via spatial and channel-wise feature alignment, several critical weaknesses and limitations should be addressed to enhance the quality and clarity of the manuscript:
1) While the proposed approach claims to enable students to "utilize their strengths and surpass teachers," there is no discussion of the added computational cost due to the upsampling and MBCA modules. An analysis of training/inference time, parameter count, and memory overhead is necessary to validate the method’s efficiency claims, especially for deployment on edge devices.
2) The paper lacks ablation studies isolating the impact of the spatial dimension alignment module and the MBCA module. Without such analysis, it is difficult to assess the individual contribution and necessity of each module in the overall framework.
3) Although the method is compared with a baseline (WRN-40-2 vs. ResNet-8x4), the paper lacks a thorough benchmarking against state-of-the-art knowledge distillation techniques (e.g., CRD, SRRL, DKD, OFD). This weakens the claim of achieving "state-of-the-art" performance without a competitive landscape.
4) The paper highlights the result where WRN-40-2 outperforms ResNet-8x4 by 2% on CIFAR-100, but does not clarify whether this is a consistent trend across other teacher–student pairs and datasets. The generalizability of this claim remains speculative without broader validation.

Experimental design

-

Validity of the findings

-

Cite this review as

---

## Round 0.2 · accepted · Accept

Thank you for your thorough revision and thoughtful responses to reviewer feedback. I am pleased to inform you that your manuscript is now accepted for publication.

In this second round, Reviewer 1 has confirmed that all previous concerns have been addressed and supports acceptance. While they suggest citing additional recent works [1,2] to further strengthen the related work section, this is not a requirement for acceptance at this stage.

Congratulations on your contribution. We look forward to publishing your work.

Reviewer 1 ·

Basic reporting

After reading the authors' response and the other reviewers' comments, the authors' revision addresses my concerns. I strongly recommand this paper could discuss and cite more recent works [1,2] published at 2025. I do not have other concerns and this paper could be accepted.

[1] Lee J, Das D, Hayat M, et al. Customkd: Customizing large vision foundation for edge model improvement via knowledge distillation[C]//Proceedings of the Computer Vision and Pattern Recognition Conference. 2025: 25176-25186.
[2] Yang C, Yu X, Yang H, et al. Multi-teacher knowledge distillation with reinforcement learning for visual recognition[C]//Proceedings of the AAAI Conference on Artificial Intelligence. 2025, 39(9): 9148-9156.

Experimental design

no comment

Validity of the findings

no comment

Additional comments

no comment

Cite this review as